

# Intrinsically disordered caldesmon binds calmodulin via the "buttons on a string" mechanism

Sergei E. Permyakov[1], Eugene A. Permyakov[1] and Vladimir N. Uversky[1,2]

[1] Protein Research Group, Institute for Biological Instrumentation, Russian Academy of Sciences, Pushchino, Moscow Region, Russia
[2] Department of Molecular Medicine, University of South Florida, Tampa, FL, USA

## ABSTRACT

We show here that chicken gizzard caldesmon (CaD) and its C-terminal domain (residues 636–771, $CaD_{136}$) are intrinsically disordered proteins. The computational and experimental analyses of the wild type $CaD_{136}$ and series of its single tryptophan mutants (W674A, W707A, and W737A) and a double tryptophan mutant (W674A/W707A) suggested that although the interaction of $CaD_{136}$ with calmodulin (CaM) can be driven by the non-specific electrostatic attraction between these oppositely charged molecules, the specificity of $CaD_{136}$-CaM binding is likely to be determined by the specific packing of important $CaD_{136}$ tryptophan residues at the $CaD_{136}$-CaM interface. It is suggested that this interaction can be described as the "buttons on a charged string" model, where the electrostatic attraction between the intrinsically disordered $CaD_{136}$ and the CaM is solidified in a "snapping buttons" manner by specific packing of the $CaD_{136}$ "pliable buttons" (which are the short segments of fluctuating local structure condensed around the tryptophan residues) at the $CaD_{136}$-CaM interface. Our data also show that all three "buttons" are important for binding, since mutation of any of the tryptophans affects $CaD_{136}$-CaM binding and since $CaD_{136}$ remains CaM-buttoned even when two of the three tryptophans are mutated to alanines.

## INTRODUCTION

Caldesmon (CaD) is a ubiquitous actin-binding protein of ∼770 residues with the molecular mass of 88.75 kDa and *pI* of 5.56 (*Mabuchi et al., 1996*). CaD is involved in the regulation of smooth muscle contraction, non-muscle motility, and cytoskeleton formation (*Czurylo & Kulikova, 2012*; *Gusev, 2001*; *Marston & Redwood, 1991*; *Martson & Huber, 1996*; *Matsumura & Yamashiro, 1993*; *Sobue & Sellers, 1991*). Particularly, CaD plays a role in a thin-filament-linked regulation of smooth muscle contraction through specific binding to F-actin and F-actin-tropomyosin leading to the inhibition of the actin-stimulated myosin ATPase (*Marston & Redwood, 1991*). The inhibitory action of CaD is reversed by interaction of this protein with various calcium-dependent proteins, such as calmodulin (CaM), caltropin (*Mani & Kay, 1996*), S100 proteins (*Polyakov et al., 1998*) and calcyclin (*Kuznicki*

Corresponding author
Vladimir N. Uversky,
vuversky@health.usf.edu

*& Filipek, 1987*). The functional activity of CaD is further regulated by phosphorylation at multiple sites (*Shirinsky, Vorotnikov & Gusev, 1999*). CaD is also engaged in the interaction with F-actin (*Adelstein & Eisenberg, 1980*; *Gusev, 2001*). These thin filament-based modulatory effects provide additional "fine-tuning" to the well-established, myosin light chain phosphorylation-dependent, thick filament-based regulation of smooth muscle contraction (*Adelstein & Eisenberg, 1980*). CaD is found to form tight complexes with several proteins, such as myosin, actin, CaM (*Marston & Redwood, 1991*), caltropin (*Gusev, 2001*; *Mani & Kay, 1996*), calcyclin (*Kuznicki & Filipek, 1987*), S100a$_o$, S100a and S100b proteins (*Polyakov et al., 1998*), and non-muscle tropomyosin (*Gusev, 2001*). It also possesses distinctive phospholipid-binding properties (*Czurylo, Zborowski & Dabrowska, 1993*; *Makowski et al., 1997*; *Vorotnikov, Bogatcheva & Gusev, 1992*; *Vorotnikov & Gusev, 1990*).

Sequence of CaD can be divided to four independent functional domains. The first N-terminal domain interacts with myosin and tropomyosin. The second domain is characteristic for smooth muscle CaD and also participates in the tropomyosin binding. The third domain is involved in the CaD interaction of with myosin, tropomyosin, and actin. The fourth C-terminal domain plays the most important role in the function of CaD, interacting with actin, various $Ca^{2+}$-binding proteins, myosin, tropomyosin, and phospholipids (*Gusev, 2001*). Furthermore, interaction of CaD with actin, tropomyosin, and CaM involves multiple sites (*Fraser et al., 1997*; *Gusev, 2001*; *Huber et al., 1996*; *Medvedeva et al., 1997*; *Wang et al., 1997*), with CaD being wrapped around its partners (*Gusev, 2001*; *Permyakov et al., 2003*).

CaD exists as two isoforms that are generated by alternative splicing of a single mRNA transcript. These CaD isoforms are differently distributed among tissues (*Abrams et al., 2012*; *Kordowska, Huang & Wang, 2006*). The light (or low molecular weight) isoform (l-CaD) is expressed in most cell types, including at low levels in smooth muscle, where it mediates actin and non-muscle myosin interaction in the cortical cytoskeleton (*Helfman et al., 1999*). The heavy (or high molecular weight) isoform (h-CaD) is expressed specifically in smooth muscle. It is believed that this isoform is capable of simultaneous binding to smooth muscle actin and myosin filaments due to the presence of a peptide spacer domain in the middle of the protein (*Wang et al., 1991*).

Based on these functional peculiarities (the ability to interact with multiple binding partners, the presence of numerous sites of posttranslational modifications, the capability to be engaged in wrapping interactions, and the presence of multiple alternatively spliced isoforms) one could conclude that CaD belongs to the realm of the intrinsically disordered proteins (IDPs), which were recognized quite recently (*Dunker et al., 2001*; *Dunker et al., 2008a*; *Dunker et al., 2008b*; *Dyson & Wright, 2005*; *Tompa, 2002*; *Uversky, 2002a*; *Uversky, 2002b*; *Uversky, 2010*; *Uversky & Dunker, 2010*; *Uversky, Gillespie & Fink, 2000*; *Wright & Dyson, 1999*) as important biologically active proteins without unique 3D-structures that represent a crucial extension of the protein kingdom (*Dunker et al., 2008a*; *Dyson, 2011*; *Tompa, 2012*; *Turoverov, Kuznetsova & Uversky, 2010*; *Uversky, 2002a*; *Uversky, 2003*; *Uversky, 2013a*; *Wright & Dyson, 1999*). IDPs and hybrid proteins containing both ordered and intrinsically disordered domains/regions (*Dunker et al., 2013*) are very

common in nature (*Dunker et al., 2000*; *Tokuriki et al., 2009*; *Uversky, 2010*; *Ward et al., 2004*; *Xue, Dunker & Uversky, 2012*; *Xue et al., 2010b*). They constitute significant fractions of all known proteomes, where the overall amount of disorder in proteins increases from bacteria to archaea to eukaryota, and over a half of the eukaryotic proteins are predicted to possess long IDP regions (IDPRs) (*Dunker et al., 2000*; *Oldfield et al., 2005a*; *Uversky, 2010*; *Ward et al., 2004*; *Xue, Dunker & Uversky, 2012*). Due to the lack of unique 3D-structures, IDPs/IDPRs carry out numerous crucial biological functions (such as signaling, regulation, and recognition) (*Daughdrill et al., 2005*; *Dunker et al., 2002a*; *Dunker, Brown & Obradovic, 2002*; *Dunker et al., 2005*; *Dunker et al., 1998*; *Dunker et al., 2001*; *Dyson & Wright, 2005*; *Tompa, 2002*; *Tompa, 2005*; *Tompa & Csermely, 2004*; *Tompa, Szasz & Buday, 2005*; *Uversky, 2002a*; *Uversky, 2002b*; *Uversky, 2003*; *Uversky, 2010*; *Uversky, Gillespie & Fink, 2000*; *Uversky, Oldfield & Dunker, 2005*; *Vucetic et al., 2007*; *Wright & Dyson, 1999*; *Xie et al., 2007a*; *Xie et al., 2007b*) that complement functions of ordered proteins (*Vucetic et al., 2007*; *Xie et al., 2007a*; *Xie et al., 2007b*) Furthermore, many IDPs/IDPRs are associated with the variety of human diseases (*Uversky et al., 2014*; *Uversky, Oldfield & Dunker, 2008*).

In our previous study, we showed that the C-terminal domain of chicken gizzard CaD, $CaD_{136}$ (636–771 fragment), is a typical extended IDP characterized by the almost complete lack of secondary structure, absence of a globular core, and a large hydrodynamic volume (*Permyakov et al., 2003*). Although $CaD_{136}$ can effectively bind to the $Ca^{2+}$-loaded CaM, this protein was shown to remain mostly unfolded within its complex with CaM (*Permyakov et al., 2003*). In this paper, we first performed comprehensive computational characterization of chicken gizzard CaD to confirm the overall disorder status of this protein. Then, we found that the $CaD_{136}$ has three major disorder-based potential binding sites located around the tryptophan residues W674, W707, and W737. To verify the role of these sites in $CaD_{136}$ binding to CaM, we designed and characterized biophysically three single tryptophan mutants (W674A, W707A, and W737A) and a double tryptophan mutant (W674A/W707A). This analysis suggests that $CaD_{136}$ potentially binds CaM via the "buttons on a charged string" mechanism. Some biological significance of these observations is discussed.

## MATERIALS AND METHODS

### Materials

Samples of chicken gizzard CaM, $CaD_{136}$, its single tryptophan mutants (W674A, W707A, and W737A), and a double tryptophan mutant (W674A/W707A) were a kind gift from Dr. Yuji Kobajashi (Department of Physical Chemistry, Institute of Protein Research, Osaka University, Osaka 565, Japan).

All chemicals were of analytical grade from Fisher Chemicals. Concentrations of CaD and CaM were estimated spectrophotometrically. Molar extinction coefficient for CaM was calculated based upon amino acids content according to *Pace et al. (1995)*: $\varepsilon_{280\,nm} = 2,980\ M^{-1}\ cm^{-1}$. For the wild type CaD $\varepsilon_{280\,nm} = 17,990\ M^{-1}\ cm^{-1}$ was used, whereas molar extinction coefficients for single and double tryptophan mutants were taken to be $\varepsilon_{280\,nm} = 12,490\ M^{-1}\ cm^{-1}$ and $\varepsilon_{280\,nm} = 6,990\ M^{-1}cm^{-1}$, respectively.

## Methods

*Absorption Spectroscopy.* Absorption spectra were measured on a spectrophotometer designed and manufactured in the Institute for Biological Instrumentation (Pushchino, Russia).

*Circular Dichroism Measurements.* Circular dichroism measurements were carried out by means of a AVIV 60DS spectropolarimeter (Lakewood, NJ., USA), using cells with a path length of 0.1 and 10.0 mm for far and near UV CD measurements, respectively. Protein concentration was kept at 0.6–0.8 mg/ml throughout all the experiments.

*Fluorescence Measurements.* Fluorescence measurements were carried out on a lab-made spectrofluorimeter main characteristics of which were described earlier (*Permyakov et al., 1977*). All spectra were corrected for spectral sensitivity of the instrument and fitted to log-normal curves (*Burstein & Emelyanenko, 1996*) using nonlinear regression analysis (*Marquardt, 1963*). The maximum positions of the spectra were obtained from the fits. The temperature inside the cell was monitored with a copper-constantan thermopile.

*Parameters of CaD136 Binding to CaM.* The apparent binding constants for complexes of calmodulin with the caldesmon mutants were evaluated from a fit of the fluorescence titration data to the specific binding scheme using nonlinear regression analysis (*Marquardt, 1963*). The binding scheme was chosen on the "simplest best fit" basis. The quality of the fit was judged by a randomness of distribution of residuals. Temperature dependence of intrinsic fluorescence was analyzed according to (*Permyakov & Burstein, 1984*).

*Differential Scanning Microcalorimetry.* Scanning microcalorimetric measurements were carried out on a DASM-4M differential scanning microcalorimeter (Institute for Biological Instrumentation of the Russian Academy of Sciences, Pushchino, Russia) in 0.48 mL cells at a 1 K/min heating rate. An extra pressure of 1.5 atm was maintained in order to prevent possible degassing of the solutions on heating. Protein concentrations were in the 0.5 to 0.7 mg/mL range. The heat sorption curves were baseline corrected by heating the measurement cells filled by the solvent only. Specific heat capacities of the proteins were calculated according to *Privalov (1979)* and *Privalov & Potekhin (1986)*.

*Sequence Analyses.* Amino acid sequences of human and chicken caldesmons (UniProt IDs: P12957 and Q05682, respectively) and human and chicken calmodulins (UniProt IDs: P62149 and P62158, respectively) were retrieved from UniProt (http://www.uniprot.org/).

The intrinsic disorder propensities of query proteins were evaluated by several per-residues disorder predictors, such as PONDR® VLXT (*Dunker et al., 2001*), PONDR® VSL2 (*Peng et al., 2005*), PONDR® VL3 (*Peng et al., 2006*), and PONDR® FIT (*Xue et al., 2010a*). Here, scores above 0.5 are considered to correspond to the disordered residues/regions. PONDR® VSL2B is one of the more accurate stand-alone disorder predictors (*Fan & Kurgan, 2014*; *Peng et al., 2005*; *Peng & Kurgan, 2012*), PONDR® VLXT is known to have high sensitivity to local sequence peculiarities and can be used for identifying disorder-based interaction sites (*Dunker et al., 2001*), whereas a metapredictor PONDR-FIT is moderately more accurate than each of the component predictors, PONDR® VLXT (*Dunker et al., 2001*), PONDR® VSL2 (*Peng et al., 2005*), PONDR® VL3 (*Peng et al., 2006*), FoldIndex (*Prilusky et al., 2005*), IUPred (*Dosztanyi et al., 2005a*),

and TopIDP (*Campen et al., 2008*). Disorder propensities of CaD and CaM were further analyzed using the MobiDB database (http://mobidb.bio.unipd.it/) (*Di Domenico et al., 2012*; *Potenza et al., 2015*) that generates consensus disorder scores based on the outputs of ten disorder predictors, such as ESpritz in its two flavors (*Walsh et al., 2012*), IUPred in its two flavors (*Dosztanyi et al., 2005a*), DisEMBL in two of its flavors (*Linding et al., 2003a*), GlobPlot (*Linding et al., 2003b*), PONDR® VSL2B (*Obradovic et al., 2005*; *Peng et al., 2006*), and JRONN (*Yang et al., 2005*).

For human CaM and CaD proteins, disorder evaluations together with the important disorder-related functional annotations were retrieved from $D^2P^2$ database (http://d2p2. pro/) (*Oates et al., 2013*). $D^2P^2$ is a database of predicted disorder that represents a community resource for pre-computed disorder predictions on a large library of proteins from completely sequenced genomes (*Oates et al., 2013*). $D^2P^2$ database uses outputs of PONDR® VLXT (*Dunker et al., 2001*), IUPred (*Dosztanyi et al., 2005a*), PONDR® VSL2B (*Obradovic et al., 2005*; *Peng et al., 2006*), PrDOS (*Ishida & Kinoshita, 2007*), ESpritz (*Walsh et al., 2012*), and PV2 (*Oates et al., 2013*). This database is further enhanced by information on the curated sites of various posttranslational modifications and on the location of predicted disorder-based potential binding sites.

Interactability of chicken CaD and CaM was evaluated by STRING (Search Tool for the Retrieval of Interacting Genes, http://string-db.org/), which is the online database resource, that provides both experimental and predicted interaction information (*Szklarczyk et al., 2011*). STRING produces the network of predicted associations for a particular group of proteins. The network nodes are proteins, whereas the edges represent the predicted or known functional associations. An edge may be drawn with up to 7 differently colored lines that represent the existence of the seven types of evidence used in predicting the associations. A red line indicates the presence of fusion evidence; a green line, neighborhood evidence; a blue line, co-occurrence evidence; a purple line, experimental evidence; a yellow line, text mining evidence; a light blue line, database evidence; a black line, co-expression evidence (*Szklarczyk et al., 2011*).

Potential disorder-based binding sites in $CaD_{136}$ (which is the C-terminal domain (636–771) of CaD) were found using three computational tools, $\alpha$-MoRF identifier (*Cheng et al., 2007*; *Oldfield et al., 2005b*), ANCHOR (*Dosztanyi, Meszaros & Simon, 2009*; *Meszaros, Simon & Dosztanyi, 2009*), and MoRFpred (*Disfani et al., 2012*). Since IDPs/IDPRs are commonly involved in protein-protein interactions (*Daughdrill et al., 2005*; *Dunker et al., 2002a*; *Dunker, Brown & Obradovic, 2002*; *Dunker et al., 2001*; *Dunker et al., 2008b*; *Dunker & Uversky, 2008*; *Oldfield et al., 2005b*; *Radivojac et al., 2007*; *Tompa, 2002*; *Uversky, 2011b*; *Uversky, 2012*; *Uversky, 2013b*; *Uversky & Dunker, 2010*; *Uversky, Oldfield & Dunker, 2005*), and since they are able to undergo at least partial disorder-to-order transitions upon binding, which is crucial for recognition, regulation, and signaling (*Dunker et al., 2001*; *Dyson & Wright, 2002*; *Dyson & Wright, 2005*; *Mohan et al., 2006*; *Oldfield et al., 2005b*; *Uversky, 2013b*; *Uversky, 2013c*; *Uversky, Gillespie & Fink, 2000*; *Vacic et al., 2007a*; *Wright & Dyson, 1999*), these proteins and regions often contain functionally important, short, order-prone motifs within the long disordered

regions. Such motifs are known as Molecular Recognition Feature (MoRF), they are able to undergo disorder-to-order transition during the binding to a specific partner, and can be identified computationally (*Cheng et al., 2007*; *Oldfield et al., 2005b*). For example, an $\alpha$-MoRF predictor indicates the presence of a relatively short, loosely structured region within a largely disordered sequence (*Oldfield et al., 2005b*), which can gain functionality upon a disorder-to-order transition induced by binding to partners (*Mohan et al., 2006*; *Vacic et al., 2007a*). In addition to MoRF identifiers, potential binding sites in disordered regions can be identified by the ANCHOR algorithm (*Dosztanyi, Meszaros & Simon, 2009*; *Meszaros, Simon & Dosztanyi, 2009*). This approach relies on the pairwise energy estimation approach developed for the general disorder prediction method IUPred (*Dosztanyi et al., 2005a*; *Dosztanyi et al., 2005b*). being based on the hypothesis that long regions of disorder contain localized potential binding sites that cannot form enough favorable intrachain interactions to fold on their own, but are likely to gain stabilizing energy by interacting with a globular protein partner (*Dosztanyi, Meszaros & Simon, 2009*; *Meszaros, Simon & Dosztanyi, 2009*). Regions of a protein suggested by the ANCHOR algorithm to have significant potential to be binding sites are the ANCHOR-indicated binding site (AIBS).

## RESULTS AND DISCUSSION

### Characterization of functional disorder in caldesmon and calmodulin

The amino acid sequences and compositions of IDPs/IDPRs are significantly different from those of ordered proteins and domains. For example, the amino acid compositions of extended IDPs/IDPRs (i.e., highly disordered proteins and regions lacking almost any residual structure (*Dunker et al., 2001*; *Uversky, 2002a*; *Uversky, 2002b*; *Uversky, 2003*; *Uversky, 2013a*; *Uversky, 2013c*; *Uversky & Dunker, 2010*; *Uversky, Gillespie & Fink, 2000*)) are characterized by high mean net charge and low mean hydropathy, being significantly depleted in order-promoting residues C, W, Y, F, H, I, L, V, and N and significantly enriched in disorder-promoting residues A, R, G, Q, S, P, E, and K (*Dunker et al., 2001*; *Radivojac et al., 2007*; *Romero et al., 2001*; *Vacic et al., 2007b*). The fractional difference in composition between CaD and a set of ordered proteins from PDB Select 25 (*Berman et al., 2000*) was calculated as $(C_{CaD} - C_{order})/C_{order}$, where $C_{CaD}$ is the content of a given amino acid in CaD, and $C_{order}$ is the corresponding value for the set of ordered proteins. This analysis revealed that in comparison with typical ordered proteins, CaD is significantly depleted in major order-promoting residues (C, Y, F, H, V, L, and I) and is significantly enriched in major disorder-promoting residues, such as A, R, E, and K. This means that CaD might contain multiple structural and functional signatures typical for the IDPs.

In agreement with this conclusion, Fig. 1A represents the results of the disorder predisposition analysis in CaD by a family of PONDR disorder predictors, PONDR® VLXT (*Dunker et al., 2001*), PONDR® VSL2 (*Peng et al., 2005*), PONDR® VL3 (*Peng et al., 2006*), and PONDR® FIT (*Xue et al., 2010a*). Since the absolute majority of residues is predicted to have disorder scores above 0.5 and since the mean disorder score for the

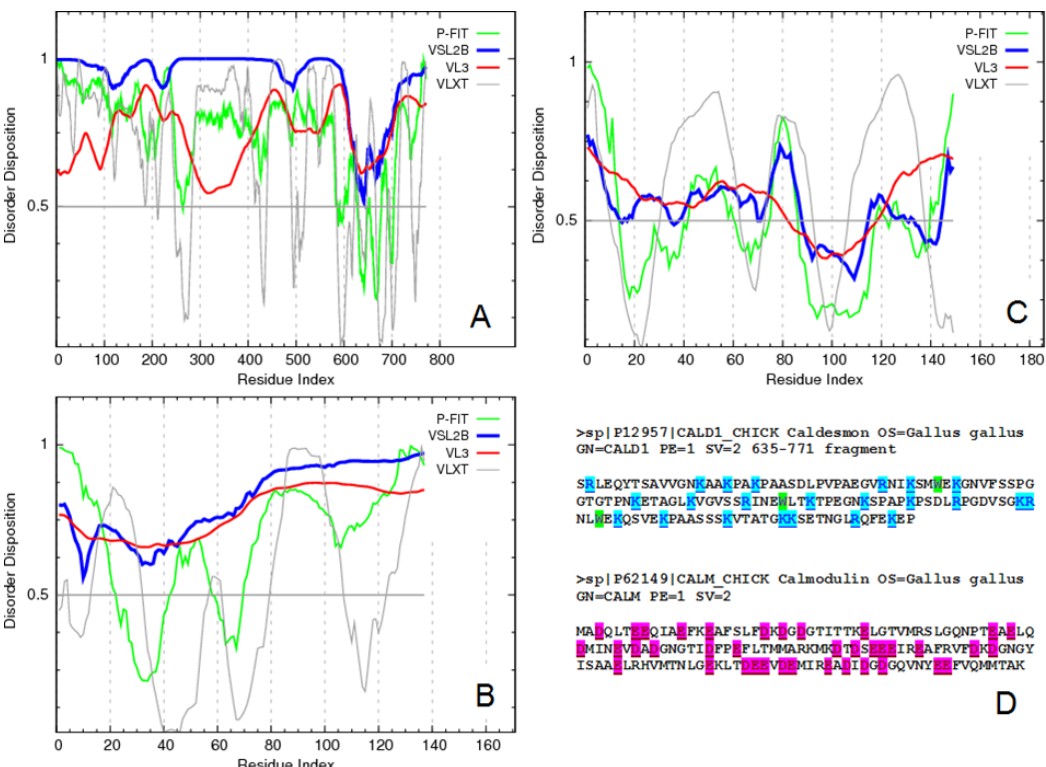

**Figure 1** Evaluating the intrinsic disorder propensities of chicken CaD (A), CaD$_{136}$ (B), and chicken CaM (C) by the family of PONDR predictors. A disorder threshold is indicated as a thin line (at score = 0.5) in all plots to show a boundary between disorder (>0.5) and order (<0.5). Plot (D) represents the amino acid sequences of CaD$_{136}$ and CaM, for which the positively and negatively charged residues are highlighted. The positions of tryptophan residues within the CaD$_{136}$ sequence are also indicated.

full-length protein ranges, depending on the predictor, from 0.69 to 0.93, this analysis clearly shows that CaD is expected to be mostly disordered. In agreement with this conclusion, the consensus MobiDB analysis (http://mobidb.bio.unipd.it/entries/P12957) revealed that chicken gizzard CaD contains 98.4% disordered residues. Curiously, the C-terminal domain of this protein, CaD$_{136}$, is predicted to be a bit more predisposed for order than the remaining protein (depending on the predictor, the mean disorder score for this 636–771 fragment of CaD ranges from 0.52 to 0.81). This observation is further illustrated by Fig. 1B which represents the PONDR-based disorder profiles of this region.

Curiously, although several X-ray crystal (PDB IDs: 1ahr, 1up5, 2bcx, 2bki, 2o5g, 2o60, 2vb6, 3gog, and 3gp2) and NMR solution structures (PDB IDs: 2kz2 and 2m3s) of CaM are known, Fig. 1C shows that this protein is predicted to be rather disordered too. These findings are not too surprising, since it is known that the CaM structure and folding are strongly dependent on the metal ion binding (*Li, Wang & Takada, 2014*; *Sulmann et al., 2014*), and that there is a great variability in the crystal structures of CaM in isolation (i.e., where it is not bound to its protein or peptide partners and exists in the unliganded form) which is considered as an illustration of CaM plasticity in solution (*Kursula, 2014*). Furthermore, several studies on the structure of unliganded CaM in

solution using small angle scattering and other methods have indicated the presence of a mixture of conformations (*Bertini et al., 2010*; *Heller, 2005*; *Kursula, 2014*; *Yamada et al., 2012*). Also in agreement with these predictions, the analysis of one of the NMR structures of CaM (PDB ID: 2m3s) revealed that this protein might contain up to 50.3% of disordered residues in solution (*Moroz et al., 2013*). Again, the results of the per-residue predictions by the members of the PONDR family are further supported by the results of the MobiDB analysis, according to which the consensus disorder content of CaM based on the outputs of ten disorder predictors is 18.1%. The corresponding values evaluated by the individual predictors (http://mobidb.bio.unipd.it/entries/P62149) are ranging from 6.0% and 13.4% for the ESpritz-XRay and DisEMBL-465, respectively to 41.6% and 69.1% for the IUPred-long and PONDR® VSL2, respectively. Note that both ESpritz-XRay and DisEMBL-465 are trained based on proteins with known crystal structures and containing regions of missing electron density, whereas IUPred-long and PONDR VSL2 use different criteria for training.

Further information on the functional disorder status of CaD and CaM was retrieved from $D^2P^2$ portal, which represents a database of pre-computed disorder predictions for a large library of proteins from completely sequenced genomes (*Oates et al., 2013*), which in addition to outputs of nine disorder predictors provides information on the curated sites of various posttranslational modifications and on the location of predicted disorder-based potential binding sites. Since this database does not include data for chicken, the human homologues of CaD and CaM were used for this analysis. The validity of this approach is justified by the fact that sequences of human and chicken CaMs are identical (100% identity), whereas sequences of human and chicken CaD are highly conserved (61% identity).

Figures 2A and 3A represents the results of this analysis of CaD and CaM, respectively, and provide further support for the abundance and functional importance of intrinsic disorder in these proteins, which are predicted to contain long disordered regions enriched in potential disorder-based binding motifs and containing numerous predicted sites of potential posttranslational modifications (PTMs). The fact that disordered domains/regions of the human CaD and CaM contain numerous PTM sites is in agreement with the well-known notion that phosphorylation (*Iakoucheva et al., 2004*) and many other enzymatically catalyzed PTMs are preferentially located within the IDPRs (*Pejaver et al., 2014*).

The interactivity of chicken CaD and CaM was evaluated by the online database resource, STRING, which provides information on both experimental and predicted interactions (*Szklarczyk et al., 2011*). Figures 2B and 3B clearly show that both proteins are predicted to have numerous binding partners. Predicted here high levels of connectivity and binding promiscuity indicate that, in the related protein-protein interaction networks (PPI), chicken CaD and CaM serve as hub proteins connecting biological modules to each other. The binding promiscuity of hub proteins is believed to be dependent on intrinsic disorder (*Dosztanyi et al., 2006*; *Ekman et al., 2006*; *Haynes et al., 2006*; *Patil & Nakamura, 2006*; *Singh et al., 2006*; *Uversky, Oldfield & Dunker, 2005*), where disorder and related disorder-to-order transitions enable one protein to interact with multiple partners (one-to-many signaling) or enable multiple partners to bind to one protein (many-to-one

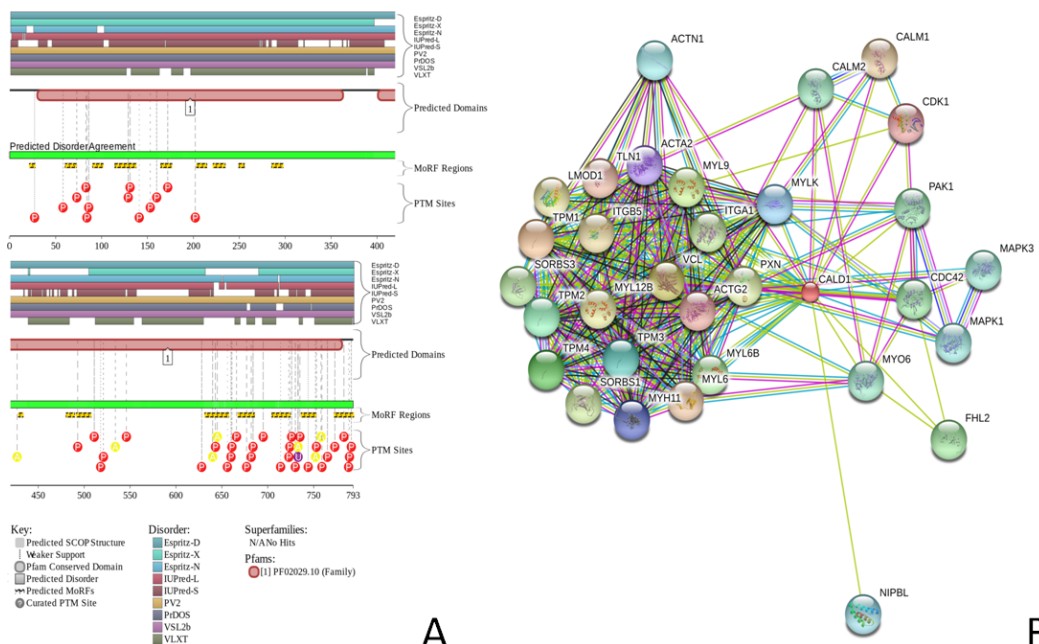

**Figure 2** Evaluation of the functional intrinsic disorder propensity of the human CaD (UniProt ID: Q05682) by the D²P² platform (http://d2p2.pro/) (*Oates et al., 2013*). In this plot, top nine colored bars represent locations of disordered regions predicted by different computational tools (Espritz-D, Espritz-N, Espritz-X, IUPred-L, IUPred-S, PV2, PrDOS, PONDR® VSL2b, and PONDR® VLXT, see keys for the corresponding color codes). Dark red bar shows the location of the functional domain found by the Pfam platform, which is a database of protein families that includes their annotations and multiple sequence alignments generated using hidden Markov models (*Berman et al., 2000; Finn et al., 2006; Finn et al., 2008*). The green-and-white bar in the middle of the plot shows the predicted disorder agreement between these nine predictors, with green parts corresponding to disordered regions by consensus. Red, yellow and purple circles at the bottom of the plot show the locations of phosphorylation, acetylation and ubiquitination sites, respectively. (B) Analysis of the interactivity of the chicken gizzard CaD (UniProt ID: P12957) by STRING (*Szklarczyk et al., 2011*). STRING produces the network of predicted associations for a particular group of proteins. The network nodes are proteins, whereas the edges represent the predicted or known functional associations. An edge may be drawn with up to 7 differently colored lines that represent the existence of the seven types of evidence used in predicting the associations. A red line indicates the presence of fusion evidence; a green line, neighborhood evidence; a blue line, co-occurrence evidence; a purple line, experimental evidence; a yellow line, text mining evidence; a light blue line, database evidence; a black line, co-expression evidence (*Szklarczyk et al., 2011*).

signaling) (*Dunker et al., 1998*). In line with these considerations, intrinsically disordered nature of chicken CaD and CaM provides a plausible explanation for their potential roles as hub proteins. Therefore, data reported in Figs. 1, 2 and 3 suggest that both CaD and CaM are expected to contain substantial amounts of functional disorder, which CaD being predicted to be mostly disordered.

Figure 1D shows that the positively charged R and K residues are evenly distributed within the CaD$_{136}$ sequence and that the sequence of CaM contains evenly spread negatively charged residues D and E. Since under the physiologic conditions of neutral pH, the C-terminal interacting domain of CaD and CaM possess charges of opposite sign (+9 for CaD$_{136}$ and -24 for CaM) it is likely that electrostatic interactions play important role in interaction between these two proteins. This hypothesis is further supported by

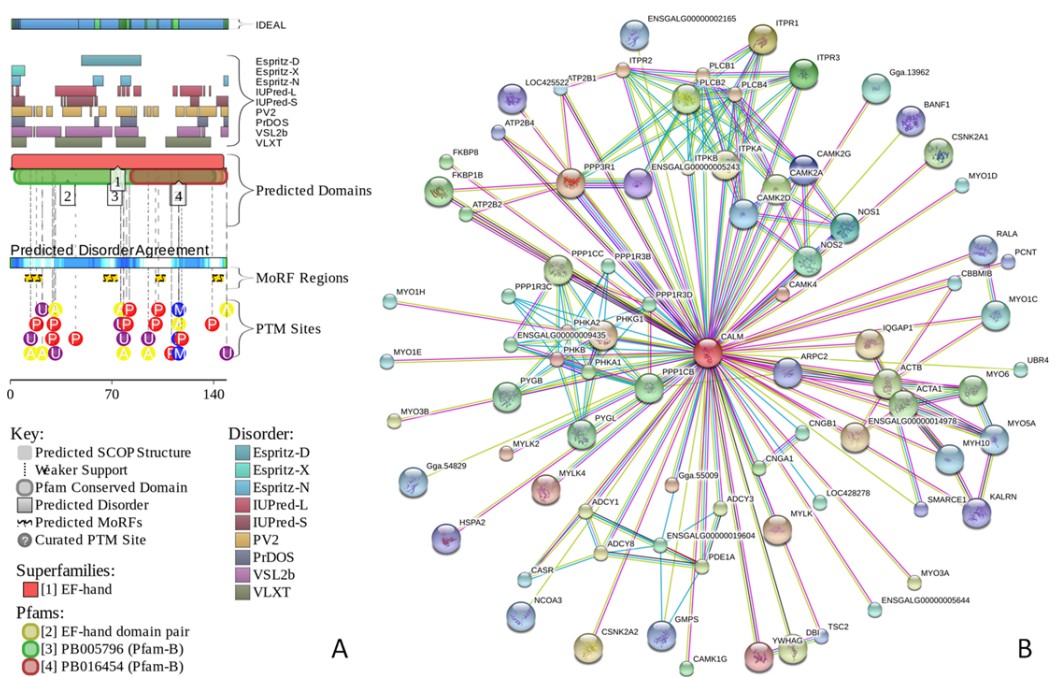

**Figure 3 Evaluation of the functional intrinsic disorder propensity of human CaM (UniProt ID: P62158) by D²P² database (http://d2p2.pro/) (Oates et al., 2013).** In this plot, top dark blue bar with green stripes shows the localization of disordered region annotated in the IDEAL database (Fukuchi et al. 2012) for this protein. Next nine colored bars represent location of disordered regions predicted by different disorder predictors (Espritz-D, Espritz-N, Espritz-X, IUPred-L, IUPred-S, PV2, PrDOS, PONDR® VSL2b, and PONDR® VLXT, see keys for the corresponding color codes). Dark red bar shows the location of the functional domain found by the Pfam platform, which is a database of protein families that includes their annotations and multiple sequence alignments generated using hidden Markov models (*Berman et al., 2000*; *Finn et al., 2006*; *Finn et al., 2008*). Blue-and-white bar in the middle of the plot shows the predicted disorder agreement between these nine predictors, with green parts corresponding to disordered regions by consensus. Red, yellow, purple and blue circles at the bottom of the plot show the location of phosphorylation, acetylation, ubiquitination, and methylation sites, respectively. (B) Analysis of the interactivity of the chicken CaM (UniProt ID: P62149) by STRING (*Szklarczyk et al., 2011*).

Fig. 4, which represents the charge distribution over the CaM surface and shows that negative charges are almost evenly distributed over the entire protein surface. What then defines the specificity of interaction between a highly positively charged IDP (CaD₁₃₆) and a highly negatively charged surface of CaM? Some answers to this important question can be obtained analyzing peculiarities of the disorder distribution in CaD$_{136}$. In fact, many IDPs/IDPRs involved in protein-protein interactions and molecular recognitions are able to undergo at least partial disorder-to-order transitions upon binding (*Daughdrill et al., 2005*; *Dunker et al., 2002a*; *Dunker, Brown & Obradovic, 2002*; *Dunker et al., 2001*; *Dunker et al., 2008b*; *Dunker & Uversky, 2008*; *Dyson & Wright, 2002*; *Dyson & Wright, 2005*; *Mohan et al., 2006*; *Oldfield et al., 2005b*; *Radivojac et al., 2007*; *Tompa, 2002*; *Uversky, 2011b*; *Uversky, 2012*; *Uversky, 2013b*; *Uversky, 2013c*; *Uversky & Dunker, 2010*; *Uversky, Gillespie & Fink, 2000*; *Uversky, Oldfield & Dunker, 2005*; *Vacic et al., 2007a*; *Wright & Dyson, 1999*). Such potential disorder-based binding sites are known as Molecular Recognition Feature

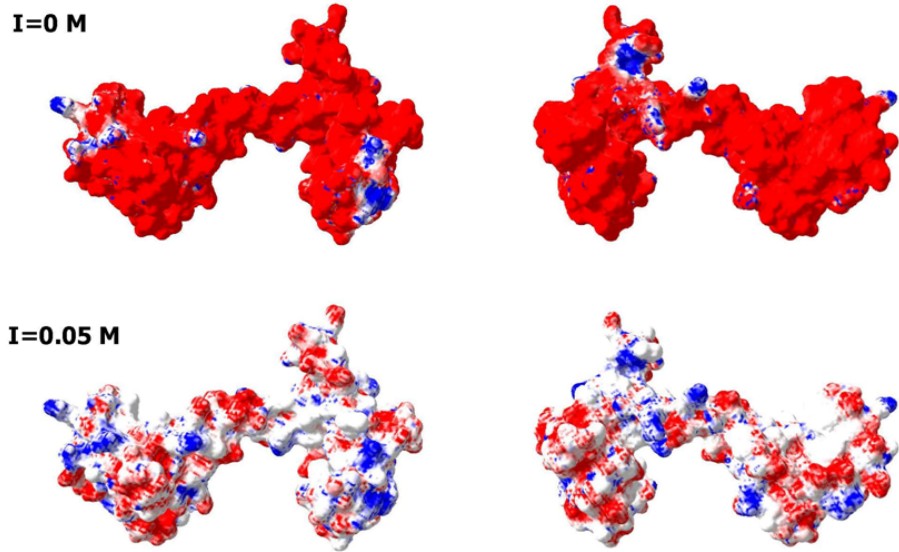

**Figure 4 Analysis of the charge distribution on the surface of CaM molecule.** PDB file: 1CLM. Analyzed protein: calmodulin, $Ca^{2+}$-form (1 chain, 4 Ca ions), without first 3 residues Ala, Gln, and Glu and without a last residue Lys. $Ca^{2+}$ ions and water molecules were removed, absent hydrogen atoms were added. Calculations were done using the Swiss-PdbViewer v3.7b2 program. Method of calculation: Poisson-Boltzmann, using partial atom charges, ionic strength 0M or 0.05M, dielectric constant of solvent 80, for protein—4. Colors: Red, potential value is NEGATIVE, $-1.8$ kT/e; White, potential value is ZERO; Blue, potential value is POSITIVE, 1.8 kT/e.

(MoRF), and they often can be found based on the peculiar shape of a disorder profile (sharp "dips" within the long IDPRs). These observations serve as a foundation for the corresponding computational tools, e.g., $\alpha$-MoRF-Pred (*Cheng et al., 2007*; *Oldfield et al., 2005b*) or MoRFpred (*Disfani et al., 2012*). Alternatively, the disorder-based binding sites can be identified by ANCHOR (*Dosztanyi, Meszaros & Simon, 2009*; *Meszaros, Simon & Dosztanyi, 2009*) (see Materials and Methods). There is generally a good agreement between the results of binding sites prediction by these two tools.

These analyses revealed that $CaD_{136}$ has several disorder-based potential binding sites and three of them correspond to the major minima in the $CaD_{136}$ disorder plots obtained by both PONDR® VLXT and PONDR-FIT (see Fig. 5). Since each of these three dip-centered potential binding sites include a tryptophan residue, we decided to mutate those tryptophans in order to evaluate their roles in the $CaD_{136}$ binding to CaM. At the first stage, the disorder propensities of three single tryptophan mutants (W674A, W707A, and W737A) and a double tryptophan mutant (W674A/W707A) were compared using PONDR® VLXT and PONDR FIT algorithms. Figure 5 represents the results of these analyses and shows that the local disorder propensities were noticeably affected by single mutations W674A and W707A and by the W674A/W707A double mutation, whereas W737A had a very minimal effect on the $CaD_{136}$ disorder profile. Although the depth of corresponding disorder minima was affected by mutations, none of these tryptophan-to-alanine substitutions completely eliminated dips. These data suggested that binding affinity of $CaD_{136}$ can be moderately affected by single substitutions W674A and

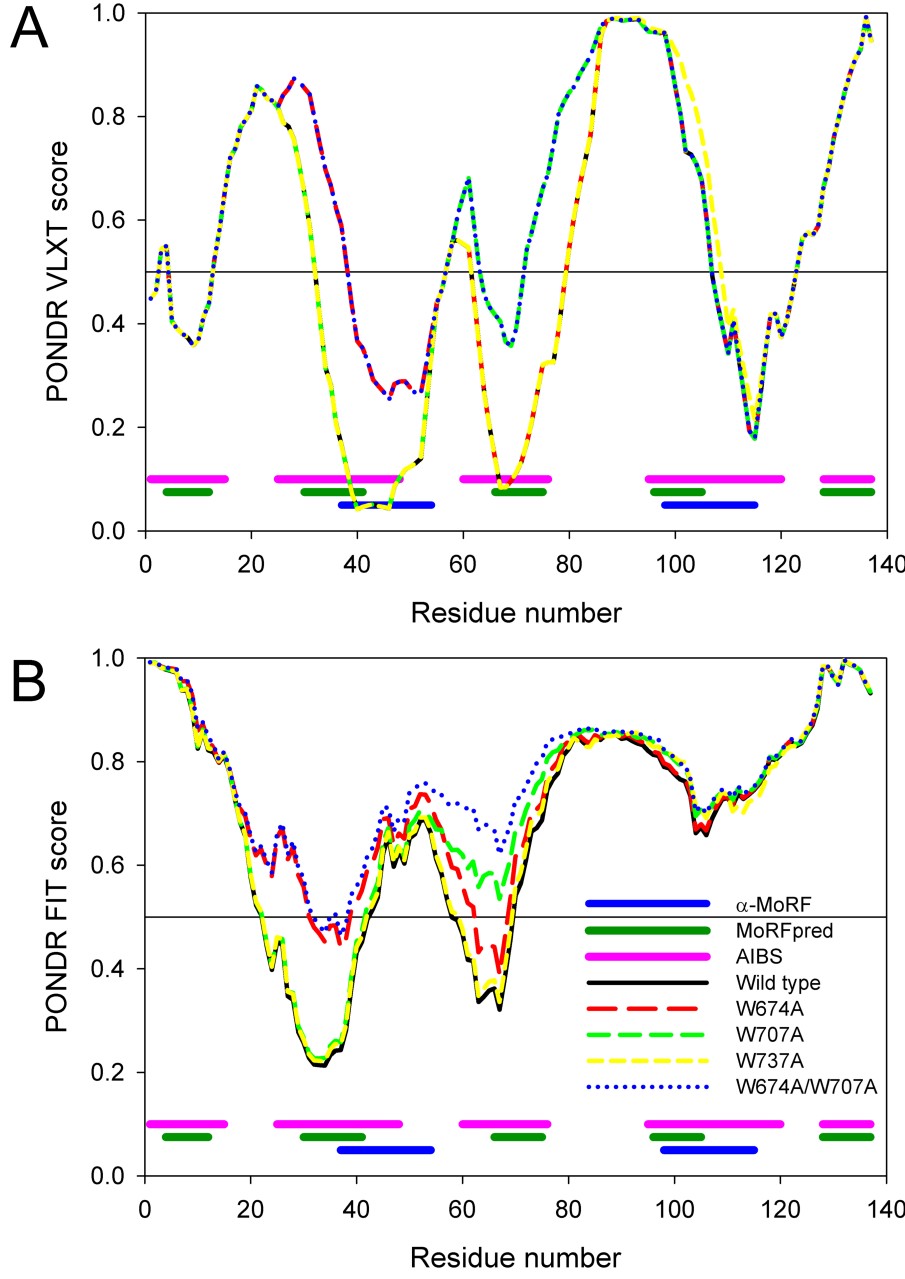

**Figure 5** **Computational analysis of the effect of tryptophan mutations on the disorder propensity of CaD$_{136}$ evaluated by PONDR® VLXT (A) and PONDR-FIT (B).** Locations of the predicted disorder-based binding sites are shown at the bottom of plots as pink (AIBSs), dark green (MoRFpreds), and dark blue (α-MoRFs) bars, respectively.

W707A, and that the W674A/W707A double mutation could have somewhat stronger effect on protein-protein interactions. To check these predictions, we analyzed biophysical properties and binding affinities of three single tryptophan mutants W674A, W707A, and W737A, and a double tryptophan mutant W674A/W707A. Results of these analyses are represented below.

**Table 1** Equilibrium association constants ($K_{CaM}$) for complexes between CaM and wild type $CaD_{136}$ and its mutants and their relative fluorescence quantum yields in the free and CaM-bound states.

| Protein | $K_{CaM}$ | $Q/Q_{trp}$ (in solution) | $Q/Q_{trp}$ (in complex with calmodulin) |
|---|---|---|---|
| WT | $(6.5 \pm 1.6) \times 10^5$ | 1.25 | 2.40 |
| W674A | $(2.2 \pm 0.6) \times 10^5$ | 1.25 | 2.72 |
| W707A | $(3.0 \pm 0.8) \times 10^5$ | 1.50 | 2.55 |
| W737A | $(1.8 \pm 0.5) \times 10^6$ | 1.49 | 2.95 |
| Double mutant | $(4.4 \pm 1.1) \times 10^4$ | 1.19 | 2.64 |

## Effect of tryptophan substitutions on tryptophan fluorescence spectrum of the C-terminal CaD domain

Analysis of the normalized tryptophan fluorescence spectra of $CD_{136}$ and its mutants in solution and in complex with CaM (which does not have tryptophan residues) revealed that the spectra of all the $CD_{136}$ proteins in their unbound forms are practically the same (see Fig. S1). They have extremely long wavelength positions and are similar to spectrum of a free tryptophan in water, which shows that in all these proteins, the tryptophan residues are totally exposed to water. The spectra of the complexes with CaM are different. The CaM-complexes W737A mutant has the most blue-shifted spectrum, whereas the W707A mutant in its bound state has the least blue-shifted spectrum. The Table 1 represents the relative fluorescence quantum yields for $CD_{136}$ and its mutants in solution and in the complex with CaM.

## Effect of tryptophan substitutions on far-UV CD spectra of $CaD_{136}$ mutants

Figure 6 represents the far-UV CD spectra of wild type, W674A, W707A, W737A and W674A/W707A $CaD_{136}$ and shows that all these proteins have far-UV CD spectra typical of the almost completely unfolded polypeptides. In other words, the data are consistent with the conclusion that at physiological conditions none of the $CaD_{136}$ domains has considerable amount of ordered secondary structure; i.e., they belong to the family of so-called natively unfolded proteins, which are the most disordered members of the realm of intrinsically disordered proteins. On the other hand, more detailed analysis of the far-UV CD spectrum shows that the wild type $CaD_{136}$, being mostly disordered, is still far from to be completely unfolded and preserves some residual structure (e.g., $[\theta]_{222}$ $\sim -3{,}000$ deg cm$^2$ dmol$^{-1}$, the minimum is located at 200, rather than at 196–198 nm, see Fig. 6).

Figure 6 shows that all amino acid substitutions affect the far-UV CD spectrum of the C-terminal CaD domain in a similar manner, inducing considerable decrease in the spectrum intensity around 200 nm. This is further illustrated by Fig. S2 that represents the difference spectra between the wild type $CaD_{136}$ and mutated domains and clearly shows that all the amino acid substitutions induce noticeable additional unfolding of the residual structure in the originally rather disordered protein.

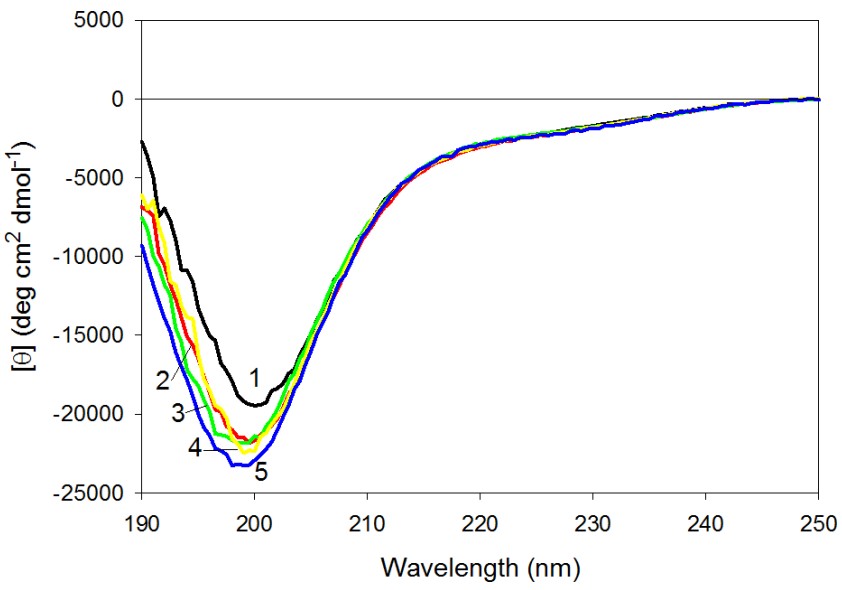

**Figure 6 Far-UV CD spectra of wild type (1), W674A (2), W707A (3), W737A (4) and W674A/W707A (5) CaD₁₃₆.** All measurements were carried out at a protein concentration of 0.6–0.8 mg/ml, cell pathlength 0.1 mm, 15 °C.

## Effect of tryptophan substitutions on the near-UV CD spectra of CaD$_{136}$ mutants

Surprisingly, Fig. 7 shows that wild type CaD$_{136}$ and all its mutants possess rather intensive and pronounced near-UV CD spectra. This means that tryptophan residues of these proteins are in relatively asymmetric environment. Figure 7 shows that any tryptophan substitution analyzed in this study has a considerable effect on the near-UV CD spectrum of CaD$_{136}$, leading to the substantial decrease in the spectral intensity. It also can be seen that different tryptophan residues have different contributions to the near-UV CD spectrum of protein. In fact, Fig. 7 shows that the effect of amino acid substitutions increases in the following order: W707A <W737A <W674A ≤ W674A/W707A. This conclusion is confirmed by the difference spectra shown in Fig. S3. Therefore, these data suggest that tryptophan residues have noticeable contributions to the residual structure of CaD$_{136}$, likely serving as condensation centers around which the local dynamic structure is formed.

## Conformational stability of CaD$_{136}$ and its mutants analyzed by the effect of temperature on their near- and far-UV CD spectra

Figure 8 represents near-UV CD spectra of the wild type and mutated CaD$_{136}$ measured at different temperatures. It can be seen that heating affects the near-UV CD spectra of different proteins in different manner. In the case of the wild type protein, some initial decrease in the spectral intensity at 40 °C is followed by the increase in spectral intensity at higher temperatures. Interestingly, after the cooling, the near-UV CD spectrum of this variant is somewhat more intensive than spectrum measured before the heating. Spectrum of W674A mutant increases with the temperature and this effect is reversible. Mutants W707A and W737A show reversible decrease in spectral intensity, whereas spectrum of

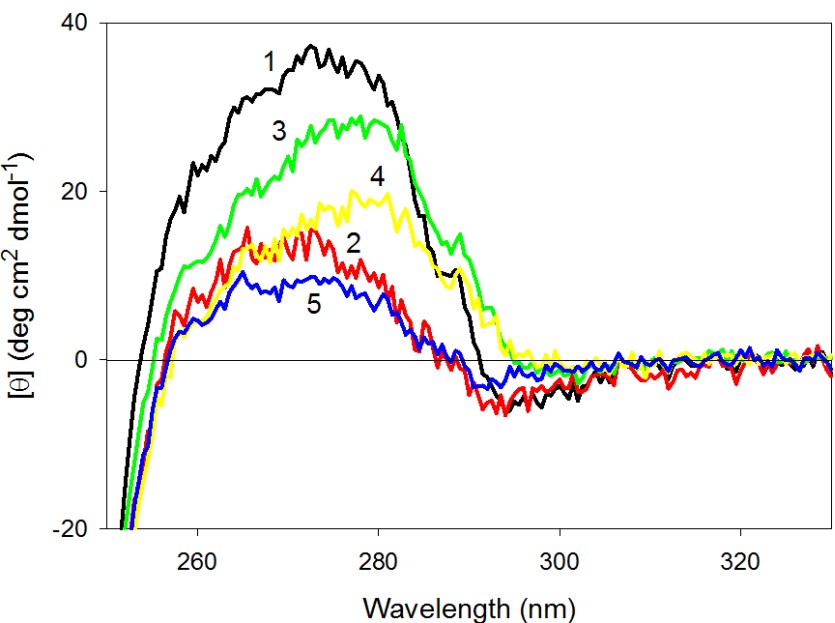

**Figure 7 Near-UV CD spectra of wild type (1), W674A (2), W707A (3), W737A (4), and W674A/W707A (5) CaD$_{136}$.** All measurements were carried out at a protein concentration of 0.6–0.8 mg/ml, cell pathlength 10 mm, 15 °C.

the double W674A/W707A mutant is practically unaffected by temperature. Importantly, Fig. 8 shows that even at 90 °C all of the protein variants analyzed in this study show pronounced near-UV CD spectra, reflecting the fact that the temperature increase does not destroy completely the asymmetric environment of their aromatic residues.

Temperature had similar effect of the far-UV CD spectra of all the CaD$_{136}$ variants. As an example, Fig. 9A represents the far-UV CD spectra of W674A mutant measured at different temperatures. It can be seen that shape and intensity of the spectrum undergo considerable changes with the increase in temperature, reflecting the temperature-induced formation of the more ordered secondary structure. Same spectral changes were observed for several other IDPs and were classified as the "turn-out" paradoxical response of extended IDPs (opposite to the response of ordered proteins) to changes in their environment (*Uversky, 2002a*; *Uversky, 2002b*; *Uversky, 2011a*; *Uversky, 2013a*; *Uversky, 2013c*; *Uversky & Dunker, 2010*). Figure 9B summarizes the data on the effect of heating on the secondary structure of the CaD$_{136}$ variants as corresponding $[\theta]_{222}$ *vs.* temperature dependences. One can see that in all cases studied temperature increase was accompanied by the steady increase in the negative ellipticity at 222 nm. It is necessary to emphasize here that this behavior is totally different from the conformational behavior of typical globular proteins, which show temperature-induced reduction in the content of ordered secondary structure.

## Studying the CaD$_{136}$ variants by scanning microcalorimetry

Figure S4 represents the calorimetric scans obtained for the wild type CaD$_{136}$ and its mutants. The absolute values of the specific heat capacity (ranging from ∼2 to 3 J/(g K))

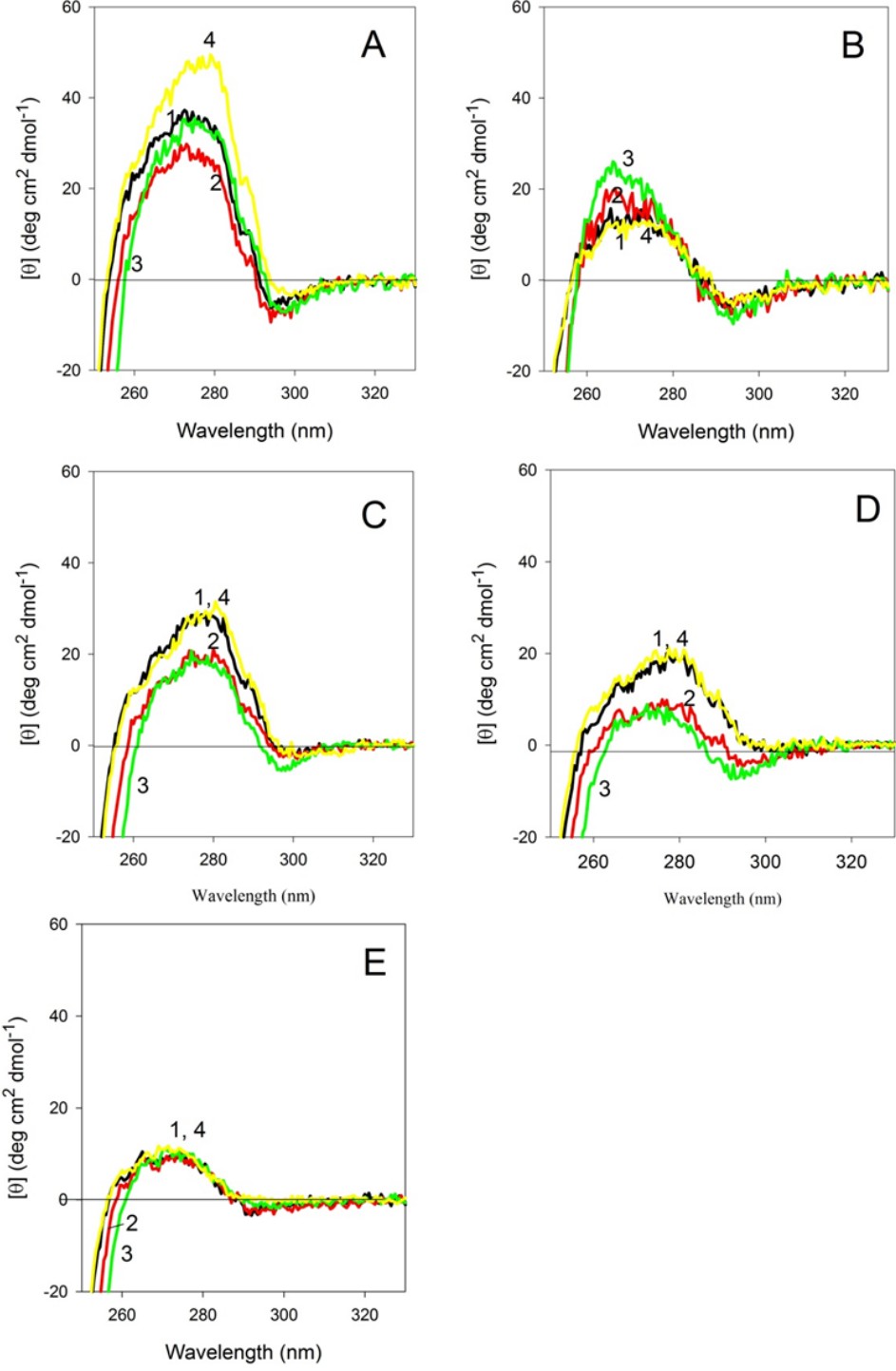

**Figure 8 Near-UV CD spectra of the wild type (A), W674A (B), W707A (C), W737A (D) and W674A/W707A (E) CaD₁₃₆ measured at different temperatures.** 15 °C (1); 40 °C (2); 90 °C (3) and 15 °C after the cooling (4). All measurements were carried out at a protein concentration of 0.6–0.8 mg/ml, cell pathlength 10 mm.

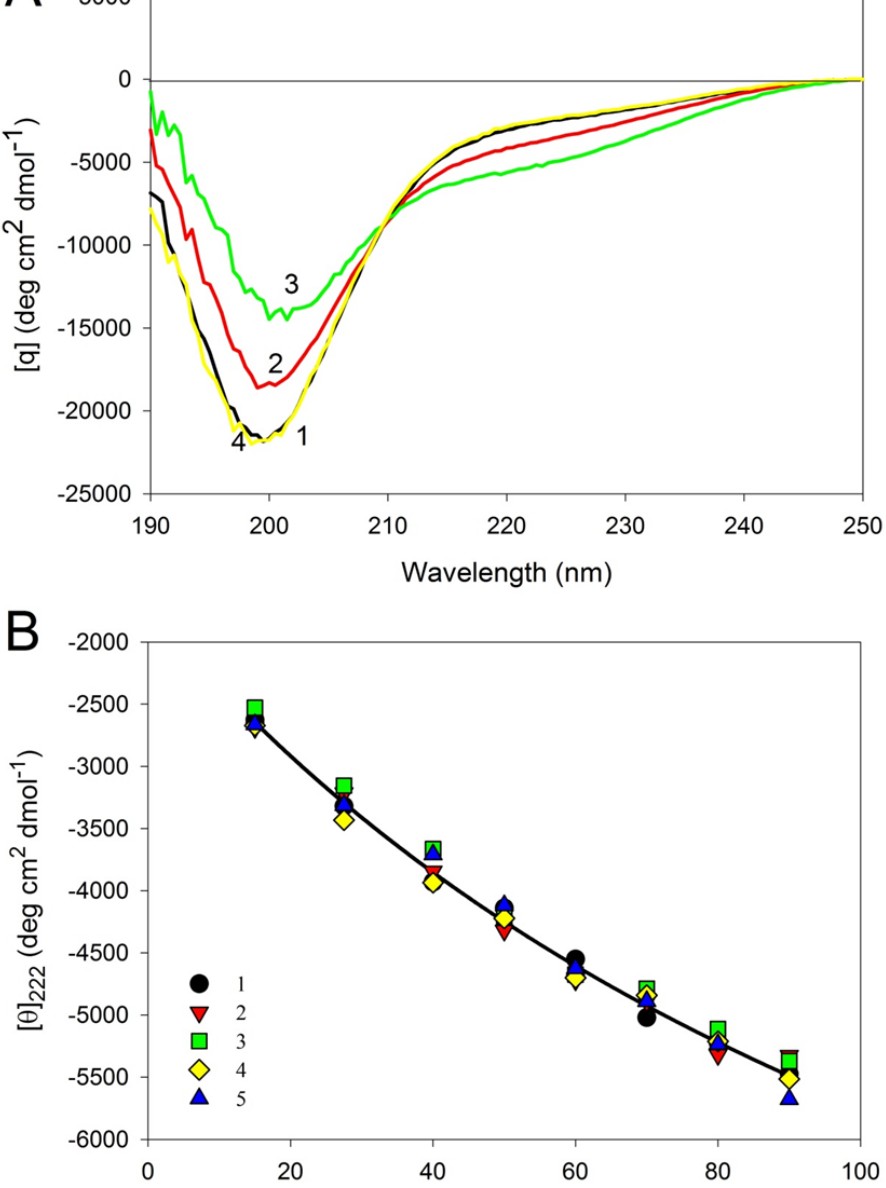

**Figure 9 Effect of temperature on far-UV CD spectra of CaD$_{136}$.** (A) Far-UV CD spectra of W674A mutant of CaD$_{136}$ measured at different temperatures: 15 °C (1); 40 °C (2), 90 °C (3) and 15 °C after the cooling (4). All measurements were carried out at a protein concentration of 0.8 mg/ml, cell pathlength 0.1 mm. (B) Effect of temperature on far-UV CD spectra of CaD$_{136}$ and its mutants: wild type (1), W674A (2), W707A (3), W737A (4) and W674A/W707A (5).

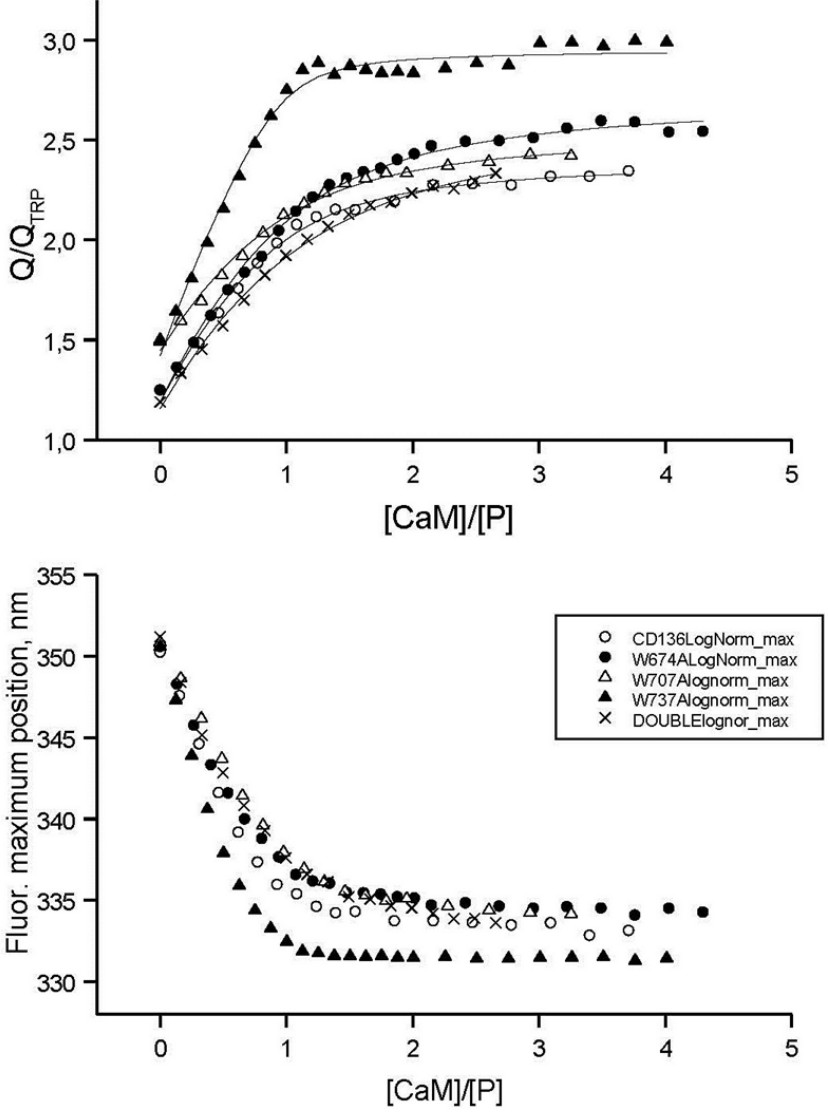

**Figure 10 Spectrofluorimetric titration of the CaD$_{136}$ and its mutants by CaM.**

and the absence of distinct heat absorption peaks within the temperature region from 10 to 100 °C for these proteins suggest that their structure is predominantly unfolded.

## Interactions of the CaD$_{136}$ and its tryptophan mutants with calmodulin studied by intrinsic fluorescence

Figure 10 represents the results of the spectrofluorimetric titration of CD$_{136}$ and its tryptophan mutants with CaM. The increase in CaM concentration induces an increase in fluorescence quantum yield and a blue shift of the fluorescence spectrum maximum (see also data presented in Fig. S1 and Table 1). The points shown in this figure are experimental data, and the curves are theoretical fits. The corresponding curves were computed using the simplest one-site binding scheme by fitting the experimental points

varying the binding constant. The values of the binding constants which give the best fits are collected in Table 1. This analysis revealed that the substitution of the tryptophan residues by alanines resulted in a decrease in the CaD$_{136}$-CaM binding constant in all the cases except W737A, where mutation caused an increase in the CaD$_{136}$ affinity for CaM. Table 1 also shows that the double W674A/W707A mutation caused the largest reduction in the CaD$_{136}$ binding efficiency. The value of the association constant for wild type CaD$_{136}$ in our work is in a good agreement with the literature data of another authors (*Czurylo et al., 1991*; *Graether et al., 1997*; *Huber et al., 1996*; *Medvedeva et al., 1997*; *Shirinsky, Bushueva & Frolova, 1988*; *Wang et al., 1997*).

The ability of the caldesmon and its C-terminal fragments to interact specifically with calmodulin has been established long ago (*Shirinsky, Bushueva & Frolova, 1988*), and several models of this complex have been suggested (reviewed in *Gusev, 2001*). It is known that the C-terminal domain of CaD contains three CaM-binding sites, centers A (close to Trp674), B (close to Trp707), and B' (close to Trp737). It has been shown that sites A and B interact with C-terminal lobe of CaM (this protein has dumbbell shape with two $\alpha$-helical Ca$^{2+}$-binding globular domains, separated by an extended "handle" formed by a seven-turn $\alpha$-helix), whereas center B forms complex with the N-terminal globular domain (*Gusev, 2001*; *Marston et al., 1994*; *Mezgueldi et al., 1994*; *Zhan, Wong & Wang, 1991*). The idea of multiple-sited interaction of CaD and CaM and participation of Trp residues in it was described earlier in a series of papers from different laboratories (for instance, *Huber et al., 1996*; *Mezgueldi et al., 1994*. For example, to determine the contribution of each of three Trp residues (659, 692, and 722, which are similar to 674, 707, and 737 in our protein) in the calmodulin-caldesmon interaction, *Graether et al. (1997)* have mutated the Trp residues to Ala in the C-terminal domain of fibroblast caldesmon (CaD39) and studied the effects on calmodulin binding by fluorescence measurements and using immobilized calmodulin (*Graether et al., 1997*). All the mutations reduced the affinity of CaD to calmodulin, but mutation of Trp 722 at site B' to Ala caused the smallest decrease in affinity. In our work similar mutation caused even an increase in affinity. The authors concluded that Trp 659 and Trp 692 are the major determinants in the fibroblast caldesmon-calmodulin interaction and that Trp 722 in site B' plays a minor role (*Graether et al., 1997*). The results of our study show that in gizzard caldesmon the letter tryptophan seems to play more significant role in the interaction with calmodulin.

## CONCLUSIONS

Altogether, data presented in our study suggest that CaD and its C-terminal domain, CaD$_{136}$, are intrinsically disordered proteins. CaD potentially serves as a disordered hub in several important protein-protein interaction networks. It is likely that CaD$_{136}$-CaM interaction is driven by the non-specific electrostatic attraction interactions due to the opposite charges of these two proteins. Specificity of CaD$_{136}$-CaM binding is likely to be determined by the definite packing of important tryptophan residues at the CaD$_{136}$-CaM interface, which is manifested by the dramatic blue shift of the intrinsic CaD$_{136}$ fluorescence. In its non-bound form, CaD$_{136}$ is highly disordered, with the

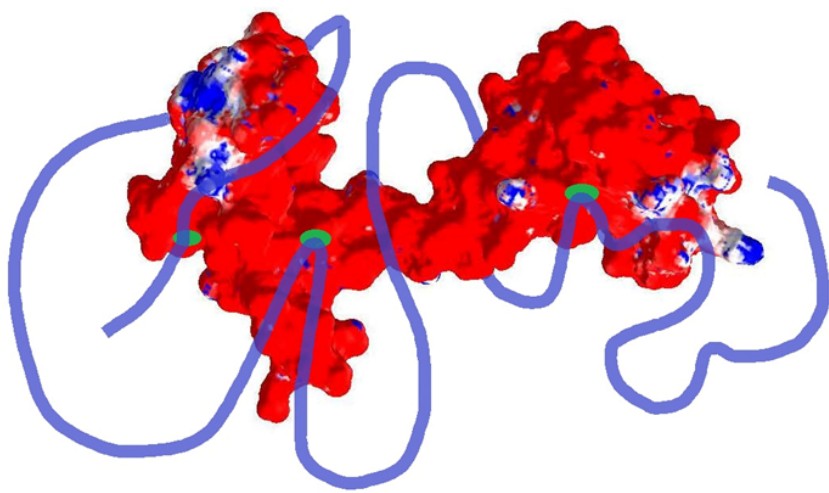

**Figure 11 Schematic representation of the "buttons on a charge string" binding mode proposed in this study.** Here, the $CaD_{136}$ is shown as a blue string containing three "buttons" (tryptophan-centric partially structured binding sites), whereas CaM is shown as mostly red surface. Note that positions of binding sites and length of the $CaD_{136}$ chain are arbitrary and used here only to illustrate an idea.

aforementioned tryptophan residues potentially serving as centers of local fluctuating structural elements. Therefore, our bioinformatics and experimental data suggest that the interaction between $CaD_{136}$ and CaM can be described within the "buttons on a charged string" model, where the electrostatic attraction between the positively charged and highly disordered $CaD_{136}$ containing at least three segments of fluctuating local structure ("pliable buttons") and the negatively charged CaM is solidified by the specific packing of three short regions containing tryptophan residues in a "snapping a button" manner. This model is schematically represented in Fig. 11. Curiously, it seems that all three "buttons" are important for binding, since mutation of any of the tryptophans affects $CaD_{136}$-CaM binding and since $CaD_{136}$ remains CaM-buttoned even when two of the three tryptophans are mutated to alanines.

## Abbreviations

| | |
|---|---|
| **AIBS** | disorder-based ANCHOR-identified binding site |
| **CaD** | caldesmon |
| **$CaD_{136}$** | C-terminal domain (636–771) of CaD |
| **CaM** | calmodulin |
| **CD** | circular dichroism |
| **DSC** | differential scanning calorimetry |
| **IDP** | intrinsically disordered protein |
| **IDPR** | intrinsically disordered protein region |
| **MoRF** | molecular recognition feature |
| **PTM** | posttranslational modification |
| **UV** | ultraviolet |

### Funding

This work was supported by grants from the Programs of the Russian Academy of Sciences "Molecular and Cellular Biology" (P.E.A.) and "Fundamental Science for Medicine" (P.S.E.). The funders had no role in study design, data collection and analysis, decision to publish, or preparation of the manuscript.

### Grant Disclosures

The following grant information was disclosed by the authors:
Programs of the Russian Academy of Sciences.

### Competing Interests

Eugene A. Permyakov and Vladimir N. Uversky are Academic Editors for PeerJ.

### Author Contributions

- Sergei E. Permyakov, Eugene A. Permyakov and Vladimir N. Uversky conceived and designed the experiments, performed the experiments, analyzed the data, wrote the paper, prepared figures and/or tables, reviewed drafts of the paper.

### Supplemental Information

Supplemental information for this article can be found online at http://dx.doi.org/10.7717/peerj.1265#supplemental-information.

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
