# Peer review of "Intrinsically disordered caldesmon binds calmodulin via the “buttons on a string” mechanism"

_PeerJ, doi:10.7717/peerj.1265_

## Round 0.1 · original submission · Minor Revisions

Please, address the reviewers' comments.

·

Basic reporting

No comments

Experimental design

No comments

Validity of the findings

No comments

Additional comments

The paper of Permyakov et al. “Does intrinsically disordered caldesmon bind calmodulin via “buttons on a string” mechanism?” deals with an interesting problem of protein-protein interaction and regulation of this process. This paper contains original experimental results and important bioinformatic data. I suppose that this paper can be recommended for publication after introduction of certain changes and addition of certain comments and explanations.

Main points.
1. The Authors claim that calmodulin (CaM) interacts with caldesmon (CaD) via mechanism of “buttons on a string” and this buttons are three Trp residues (W674, W707 and W737) located in the C-terminal domain of CaD. If this conclusion is correct then mutation of any of these residues should decrease binding of CaM to CaD. However, the point mutations of W674 and W707 practically do not affect interaction of two proteins, whereas mutation W737A unexpectedly increases interaction of CaM with the C-terminal domain of CaD. I suppose that the Authors should clearly explain this apparent discrepancy.
2. Multiple-sited interaction of CaD and CaM was described earlier in a series of papers from different laboratories (for instance, Huber et al., Biochem. J. (1996), 316, 413-420, Mezgueldi et al.,(1994) J. Biol.Chem. 269, 12824-12832). In these papers other Trp residues were postulated to serve as a buttons responsible for tight interaction of CaD and CaM. I suppose that the Authors should indicate this fact and compare their results with the earlier published data.

Minor points.
1. Lines 39-40. It is indicated that CaD is exclusively located in contractile domain of smooth muscle. At the same time later (lines 70-77) the Authors mention that CaD is expressed in most cell types and that there are at least two different isoforms of CaD.
2. Line 47. It is indicated that CaD is regulated by different Ca-binding proteins such as calmodulin and caltropin without mentioning of S100 and calcyclin which suddenly appeared later.
3. Lines 50-51. It is stated that “CaD is alternatively bound either to F-actin or to CaM”. I am afraid that this is a wrong assumption. Binding of CaM does not lead to dissociation of CaD from actin filaments.
4. Line 55-57. Interaction with tropomyosin is mentioned twice in the same sentence.
5. Line 293 and Fig.2A and 3A. The Authors present the data predicting certain post-translational modifications of CaD and CaM. I suppose that the Authors should clearly indicate that these predictions are purely theoretical and are not confirmed by experimental results.
6. Fig. 5 the numbers on abscissa indicate the residue number in CaD C-terminal fragment and therefore it is difficult to correlate these numbers with the real numbering in the full-size CaD and with the corresponding Trp residues.
7. Fig.6. The legend to this figure and the insert on the figure are not complete and do not provide enough information about the corresponding curves. I suppose that this Figure should be presented in color since otherwise it is very difficult to distinguish different curves. Normalized fluorescence intensity should be presented with decimal point, not with comma.
8. Fig.7B and Fig.8B do not contain important additional information and therefore can be omitted.
9. Fig. 11 does not present any additional information. I suppose that it is enough do describe it in the corresponding part of manuscript without presenting additional figure. I suppose that in the legend it is better to indicate that all experiments were performed in 50 mM borate buffer, but not in 50 mM H3BO3 with pH 8.0.
10. Table 1. I think that the Authors should indicate not only mean values of association constant, but also standard error or standard deviation of these values.
11. In the Reference list the Authors included abstracts (Bogatcheva N.V., Gusev N.B. J. Muscle Res. Cell Motil. (1996) 17, 146 and Vorotnikov A.V., Gusev N.B. J. Muscle Res Cell Motil. (1990a) 11, 442) which contain minimal information. I suppose that it is better to omit abstracts from the References and to cite only complete papers of the same or other authors.

·

Basic reporting

The article is mostly well written but I do have several corrections and some editorial comments:
1) The title would be better written as a definitive statement rather than a question.
2) Keywords: MoRF should be written out in full
3) Page 15 line 292 - PTMs should be in brackets
4) Page 19 line 380 - change 'muted' to 'mutated'
5) Page 20 line 416 - explain what is meant by the turn-out response
6) Page 32 line 1 - explicitly define Kcam
7) Page 35 Figure 2 legend - check that spaces are correctly printed (several of the words were run together in my file)
8) Page 46 Figures 7B and 8B - change y-axis label to Δθ (delta-theta).

Experimental design

The experimental design is for the most part clearly defined and performed at a high technical standard.

1) For Figure 6 normalized fluorescence intensity should be defined. Presumably each curve was normalized to itself since each Trp mutation would decrease the total fluorescence yield. It would be helpful to list the maximum emission wavelength of each protein in the Results.

2) The CD spectra and difference spectra in Figures 7B, 8 and 9 could benefit from a moderate smoothing algorithm to better see the trends and maxima/minima.

3) Figure 13 understandably is just an illustration, but the sense of scale could be better presented. The authors can make educated guess about the approximate distances between the three Trp residues in caldesmon, and should also show the full length of the protein. The wrapping around of parts of the caldesmon would suggest that parts of this region are weakly interacting, and should be removed.

Validity of the findings

The major comment I have is that their work is very similar to a study using the C-terminal domain of caldesmon (CaD39) and calmodulin published many years ago (Graether et al. Biochemistry 1997). This manuscript deletes the same four tryptophans in caldesmon to alanine and the same double mutant (note that the numbering system is different between the two papers) but also deletes several residues around each tryptophan. The dissociation constants were also measured for these mutants.

It would be very interesting for the present authors to comment and speculate on their present work compared to the older work, especially with regards to the deletion of several residues (some of which are charged) around two key tryptophan residues. An explanation of the increased affinity of W737A would be especially intriguing.

---

## Round 0.2 · accepted · Accept

Please, address two minor comments made by one of the reviewers. It may be possible to make these changes in production proofs.

·

Basic reporting

No comments

Experimental design

No comments

Validity of the findings

No comments

Additional comments

The Authors carefully addressed all reviewers’ comments and introduced corresponding changes in the text of manuscript.

I detected only two misprints:

Page 21, line 444. It is written “The value of the association constant for wild type CaD136 in our work is in a good agreement with the literature data of another authors…”. I suppose that it is better to write “The value of the association constant for wild type CaD136 in our work is in a good agreement with the literature data of other authors…”

Page 22, line 466. It is written ” The results of our study show that in gizzard caldesmon the letter tryptophan seems to play more significant role in the interaction with calmodulin”. I suppose that the word “letter” should be replaced by “later”.

I think that the revised version of this manuscript can be accepted for publication.

·

Basic reporting

No comments

Experimental design

No comments

Validity of the findings

No comments